posttraumatic stress disorder; PTSD; substance use; alcohol; global mental health; low-income countries; review; alcohol use disorder; comorbidity

**Author for correspondence:**
Debra Kaysen,
Email: dkaysen@stanford.edu

# Comorbid posttraumatic stress disorder and alcohol use disorder in low- and middle-income countries: A narrative review

Debra L. Kaysen[1] [iD], Katherine van Stolk-Cooke[1] [iD], Debra Kaminer[2] [iD], M. Claire Greene[3] [iD], Teresa López-Castro[4] [iD] and Jeremy C. Kane[5] [iD]

[1]Division of Public Mental Health and Population Sciences, Department of Psychiatry and Behavioral Sciences, Stanford University School of Medicine, Stanford, CA, USA; [2]Department of Psychology, University of Cape Town, Cape Town, South Africa; [3]Program on Forced Migration and Health, Heilbrunn Department of Population and Family Health, Columbia University Mailman School of Public Health, New York, NY, USA; [4]Department of Psychology, Colin Powell School of Civic and Global Leadership, The City College of New York, 160 Convent Avenue, New York, NY, USA and [5]Department of Epidemiology, Columbia University Mailman School of Public Health, New York, NY, USA

## Abstract

Much of the research on posttraumatic stress disorder (PTSD) and alcohol use disorder (AUD) has been conducted in high-income countries (HICs). However, PTSD and AUD commonly co-occur (PTSD + AUD) are both associated with high global burden of disease, and disproportionately impact those in low- and middle-income countries (LMICs). This narrative review attempts to synthesize the research on prevalence, impact, etiological models, and treatment of PTSD + AUD drawing from research conducted in HICs and discussing the research that has been conducted to date in LMICs. The review also discusses overall limitations in the field, including a lack of research on PTSD + AUD outside of HICs, issues with measurement of key constructs, and limitations in sampling strategies across comorbidity studies. Future directions are discussed, including a need for rigorous research studies conducted in LMICs that focus on both etiological mechanisms and on treatment approaches.

## Impact statement

Despite the fact that the majority of the world's population lives in low- and middle-income countries (LMICs), most of the research on the co-occurrence of posttraumatic stress disorder (PTSD) and alcohol use disorder (AUD) comes from data on samples from high-income countries. This review entailed a targeted examination of the existing literature on the prevalence, consequences, etiology, and treatment of comorbid PTSD + AUD derived from LMICs. Emphasis is placed on a need for more research on the epidemiology, etiology, and treatment of PTSD + AUD in LMICs. It further offers guidance based on the data thus far for how to conduct this research in methodologically sound, resource-sustainable, and culturally relevant ways.

Due to strong, consistent associations between posttraumatic stress disorder (PTSD) and alcohol use disorder (AUD), their co-occurrence (PTSD + AUD) has been a topic of extensive study (Pietrzak et al., 2011; Smith and Cottler, 2018). Data from 42 studies conducted primarily in the USA and other high-income countries (HICs) estimate that 10–61% of people with PTSD also misuse alcohol. Among people with AUD, an estimated 2–63% was found to have co-occurring PTSD (Debell et al., 2014). Studies examining variation in the presentation of PTSD + AUD have found that it is generally associated with more severe PTSD symptoms and higher rates of drinking relapse (McCarthy and Petrakis, 2010; Debell et al., 2014; Smith and Cottler, 2018). However, these findings are largely based on research conducted in HICs, and overall there is a dearth of research on the onset, course, and treatment of comorbid PTSD and AUD in low- and middle-income countries (LMICs; Seedat and Suliman, 2018). This is problematic, given that the majority of the world's population resides in LMICs (Jacob et al., 2007).

## Global burden of PTSD and AUD

Individually, PTSD and AUD present major public health burdens and are associated with high rates of functional impairment. Mental health and substance use disorders (SUDs) are the highest contributors to years lived with disability globally (Whiteford et al., 2013). Alcohol use is the

seventh leading risk factor for all deaths and disability, with 5.1% of the global burden of disease attributable to alcohol use (World Health Organization, 2018; Iranpour and Nakhaee, 2019). PTSD affects approximately 4% of the world's population and is estimated to contribute approximately as much to the global burden of disease as schizophrenia and more than anxiety disorders (Vos and Mathers, 2000; Ayuso-Mateos, 2002; Benjet et al., 2016).

Despite consistent and robust findings documenting the respective prevalence rates and consequences of PTSD and AUD, available data on PTSD + AUD are limited. The lack of research on PTSD and AUD in LMICs (Seedat and Suliman, 2018) presents a major limitation to our understanding of the onset and course of PTSD + AUD globally. Most of the available epidemiological literature on PTSD + AUD is from HICs, with a large proportion focused on military populations and relatively small proportion reporting data from general population surveys (Debell et al., 2014; Lo et al., 2017). Studies conducted in the USA have found that individuals with PTSD + AUD demonstrate greater psychiatric impairment, suicidal ideation, and social impairment than those with PTSD alone or AUD alone (Simpson et al., 2019), underscoring the need to better understand the onset and course of PTSD + AUD in LMICs in order to address the global burden of disease caused by the comorbidity.

## Epidemiology of PTSD and AUD in LMICs

Although the literature from LMICs is limited, the existing evidence suggests that while respective rates of PTSD and AUD appear to be higher in HICs and upper-middle income countries, the burden of harm is greater in LMICs and among those of lower socioeconomic status (Loring, 2014; Koenen et al., 2017). Alcohol consumption is positively associated with country income group, and the unconditional lifetime prevalence of AUD in LMICs (5.9%) is lower than that of HICs (7.2%; Glantz et al., 2020). However, heavy episodic drinking is highest in LMICs (Parry and Amul, 2022) and low socioeconomic status is associated with increased risk of alcohol-related death (Loring, 2014). Similarly, PTSD prevalence ranges from 2.1 to 2.3% in LMICs, compared to 5.0% in HICs (Koenen et al., 2017). Although trauma exposure risk is higher in LMICs (Atwoli et al., 2015), particularly for insidious forms of trauma such as exposure to war zones (Hoppen et al., 2021), treatment seeking for PTSD is more common in HICs (Koenen et al., 2017) and there are fewer evidence-based treatment options available to individuals living in LMICs (Patel et al., 2011).

Rates of AUD and PTSD each vary widely by country (Koenen et al., 2017; World Health Organization, 2018), highlighting the influence of cultural and socioeconomic factors on prevalence and burden. Moreover, studies of PTSD prevalence in LMIC countries tend to focus on examining rates of PTSD among high-risk populations such as refugees, disaster survivors, conflict-affected populations, or gender-based violence survivors, rather than within the general population (Horyniak et al., 2016; Lo et al., 2017). Additionally, there is substantial clinical and methodological heterogeneity in the epidemiological studies that have been conducted on PTSD and on AUD, precluding cross-study exploration of risk and protective factors. Some of this heterogeneity is explained by variable approaches to measuring AUD. Existing reviews describe the co-occurrence of PTSD and *alcohol misuse*, which is used as a term to cover a range of alcohol-related conditions from risky alcohol use behaviors to AUD (Roberts et al., 2022). Additionally, the

assessment tools developed in HIC settings used to measure PTSD and AUD are not always adequately validated or adapted for LMIC contexts (Nadkarni et al., 2019; Mughal et al., 2020). Taken together, definitional heterogeneity, the lack of diversity in study populations, and limitations to the cultural appropriateness of standardized assessments of AUD and PTSD make it difficult to determine the extent to which the epidemiological data from HICs generalize to LMICs, where drinking behavior, trauma exposure, and trauma response may vary considerably.

Although some studies in LMICs have examined the impact of AUD or of PTSD separately, literature on the epidemiology and impacts of PTSD + AUD is limited and has produced mixed findings (Lo et al., 2017). In specific, at-risk populations (e.g., those living with HIV), cross-sectional studies have found that PTSD or probable PTSD was significantly associated with current alcohol use, heavy alcohol use, or AUD commonly co-occur (Dévieux et al., 2013; Duko et al., 2020; Kekibiina et al., 2021). Studies designed to evaluate this comorbidity have identified a significant increase in the likelihood of PTSD + AUD in refugee samples in Uganda (Bapolisi et al., 2020) and male urban trauma survivors in Sri Lanka (Dorrington et al., 2014). This increased risk of PTSD + AUD has also been observed in population-based studies (e.g., in Brazil; de Castro Longo et al., 2020), with a particularly strong co-occurrence observed in rural settings (Colombia; Gaviria et al., 2016). However, one population-based study examining lifetime prevalence of mental disorders by AUD status in Colombia and two studies evaluating harmful alcohol use among internally displaced persons in Georgia and Uganda did not find relationships between PTSD and AUD (Roberts et al., 2011, 2014; Rincon-Hoyos et al., 2016). Gender as a risk factor for PTSD + AUD in LMICs may differ from those reported in studies conducted in HICs. For example, one study conducted in refugee camps in Croatia identified increased rates of PTSD + AUD among men, but not women (Kozarić-Kovačić et al., 2000). Similarly, in a study of urban trauma survivors in Sri Lanka, the relationship between PTSD and AUD was only found among men (Dorrington et al., 2014). This is in contrast to studies conducted in HICs examining gender differences in PTSD + AUD rates, which have generally found PTSD increases risk of AUD among women more so than among men (Olff et al., 2007; O'Hare et al., 2009; Kachadourian et al., 2014).

Differences in the epidemiology of PTSD + AUD may reflect limitations and heterogeneity in the evidence. Approaches to sampling and measurement across these studies vary, which may explain differences in findings. Even in circumstances where there is consistency in measurement, cross-cultural differences may make certain tools less valid in some contexts and populations, introducing further variability. Additionally, much of the literature on PTSD in LMICs focuses on civilian populations affected by humanitarian emergencies where the prevalence of PTSD and common mental disorders is generally higher than that observed in the general population (Charlson et al., 2019).

Exposure to potentially traumatic events and norms around alcohol, which are central to the etiology and assessment of PTSD and AUD, respectively, is known to vary across cultures and contexts. Together, differences in the characteristics of various populations, how they are sampled, and the validity and comparability of measurement tools used to assess AUD and PTSD may contribute to high variability in rates of PTSD + AUD observed across studies. In addition, the PTSD diagnostic criteria may not translate cross-culturally, introducing additional heterogeneity across studies (Michalopoulos et al., 2015). Further research to

disentangle this variability is needed to distinguish true differences in the epidemiology of PTSD + AUD from variation that can be explained by methodological limitations of the available literature.

## Theoretical models of PTSD + AUD

There are multiple explanatory theories for the association between PTSD and AUD, including the susceptibility model and the self-medication/negative reinforcement model (Haller and Chassin, 2014). The susceptibility model proposes that alcohol use increases risk of PTSD both through increased exposure to traumatic events and increased vulnerability to develop PTSD. Alcohol intoxication can increase vulnerability to perpetrators of interpersonal trauma and increase likelihood of engaging in risky behavior (Chilcoat and Breslau, 1998; Jacobsen et al., 2001; Brady et al., 2004; Schumm and Chard, 2012; Afzali et al., 2017). Alcohol use appears to increase risk of exposure to traumas such as accidental injury, sexual assault, and violence (Read et al., 2013; Lorenz and Ullman, 2016; Duke et al., 2018). Alcohol use may inhibit individuals' ability to respond to risk cues due to alcohol's impairing effects on perception (e.g., alcohol myopia; Broach, 2004; Griffin et al., 2010) or movement and coordination (Valenstein-Mah et al., 2015). After trauma exposure, alcohol use may also increase the likelihood to develop PTSD by inhibiting trauma processing and preventing habituation (Brady et al., 2004; Back et al., 2006; Balachandran et al., 2020). For example, alcohol use has been found to prospectively increase risk of PTSD among women who have experienced sexual or physical assault (Kaysen et al., 2011; Read et al., 2013).

The theory that has received the preponderance of both research and support is the self-medication model (Khantzian, 1997; Hawn et al., 2020). This theory rests on a negative reinforcement paradigm, wherein alcohol is consumed as a means to reduce distress related to the traumatic event or symptoms of PTSD. The short-term reduction in symptoms and distress then serves as a reinforcer for continued alcohol use. Over time, this can lead to independent alcohol use that develops into AUD (Smith and Cottler, 2018; Hawn et al., 2020). In longitudinal studies, trauma exposure and PTSD precede increases in drinking (Boscarino et al., 2011; Read et al., 2012; Haller and Chassin, 2014; Lane et al., 2019). In micro-longitudinal studies, such as studies using experience sampling or daily diary designs, which collect a large amount of data typically over moments, days, or weeks, increases in PTSD symptoms precede alcohol use (Sullivan et al., 2020; Dworkin et al., 2021). Taken together, these studies provide support for the self-medication model, highlighting potential mechanisms such as coping-related drinking, emotion regulation, and negative affect (Luciano et al., 2022; Weiss et al., 2022).

Studies in LMICs have found relationships between alcohol use and increased risk of trauma exposure, although these studies have not tested this in relation to PTSD. For example, alcohol use has been associated with risk of intimate partner violence in several studies conducted in Africa (Shamu et al., 2011; Greene et al., 2017). For relationships between PTSD and alcohol use, most studies presume self-medication as the mechanism of action. In one of the few studies looking at potential mechanisms in a LMIC, in Uganda, alcohol was found to moderate relationships between trauma exposure and PTSD, which was attributed to self-medication (Ertl et al., 2016). However, overall in LMICs little research has tested these causal models and those that have rely on cross-sectional designs, which preclude examining temporal effects.

The models of PTSD + AUD comorbidity have largely been developed in HIC settings and may not encompass other potential explanatory mechanisms (Shuai et al., 2022). Models of broad relevance to LMIC settings may further be limited by the fact that, as previously discussed, prior examinations of PTSD + AUD in LMICs have focused on specific, at-risk groups rather than population-level data. Syndemics theory has been proposed as a helpful framework for understanding PTSD + AUD associations and identifying additional avenues for intervention in some global settings (de Jong et al., 2015). Syndemics theory proposes that intersecting epidemics, such as violence, HIV, and substance use, interact synergistically to compound the burden of disease (Singer et al., 2017; Mendenhall et al., 2022). This theory considers the role of harmful social conditions in understanding disease concentration and interaction. In addition, there is a need for research examining ways in which culture, community, and local context may promote resilience, given that PTSD, AUD, and PTSD + AUD are not evenly distributed across communities (Pollack et al., 2016). The lack of research evaluating theories of PTSD + AUD outside of HIC's is especially problematic, as theories often drive treatment approaches and help to identify target mechanisms for intervention.

## Treatment of PTSD + AUD in HICs

Treatment for PTSD + AUD has overwhelmingly been designed in and for HIC settings and has undergone significant shifts over time, influenced by improvements in best practice guidelines for the treatment of each respective disorder and evolutions in conceptualizations of PTSD + AUD itself. The contemporary repertoire of evidence-based AUD care includes cognitive-behavioral treatments (CBTs) such as relapse prevention and mindfulness-based relapse prevention, motivational enhancement therapy, family/couples interventions (e.g., the community reinforcement approach), and pharmacotherapies (e.g., disulfiram, naltrexone, and acamprosate). For PTSD, trauma-focused psychotherapies have garnered the strongest empirical support and are broadly recommended as the front-line interventions (Bisson et al., 2019; Lewis et al., 2020). The umbrella term of *trauma-focused therapies* refers to treatments that involve the direct processing of trauma-related cues, including memories, thoughts, physiological sensations, and external reminders. Prevailing trauma-focused therapies include prolonged exposure, cognitive processing therapy, trauma-focused CBT, and eye movement desensitization and reprocessing (EMDR).

Historically, the dominant philosophy for treating PTSD + AUD advised applying AUD and PTSD treatments sequentially; alcohol-related problems were treated first, aiming to achieve a state of clinical stability (usually defined as a period of alcohol abstinence), and then PTSD was addressed (Back et al., 2019). This *sequential* approach was largely predicated on clinical concerns that prematurely employing trauma-focused therapies would exacerbate AUD symptoms and/or derail recovery gains in non-abstinent individuals (Becker et al., 2004).

Beginning in the early 2000s, scientific advances in the study of the functional relationship between PTSD and AUD suggested that addressing trauma symptoms could facilitate reductions in alcohol use by reducing the need for coping-related drinking (Coffey et al., 2002). At the same time, a growing interest in non-abstinence outcomes and harm reduction approaches led to the widening of AUD treatment goals to include moderation alongside complete abstinence (U.S. Department of Health and Human Services, 2015). Against this backdrop, a new wave of *concurrent* treatments

emerged that target PTSD + AUD simultaneously. Concurrent approaches either provide stand-alone PTSD and AUD treatments conducted in parallel, or integrated interventions in which PTSD + AUD is treated by the same clinician (e.g., *Concurrent Treatment of PTSD and Substance Use Disorders with Prolonged Exposure*, Back et al., 2019). Clinical trials data of concurrent, trauma-focused PTSD + AUD treatments have consistently shown reductions in PTSD without worsening of AUD (Roberts et al., 2022). Crucially, these studies have suggested that improvements in PTSD also lead to future decreases in AUD symptoms, but that the inverse is much less likely (Hien et al., 2018).

Several conventional meta-analyses and systematic reviews including data from HICs now generally point to the superiority of concurrent trauma-focused interventions over sequential treatments or treatments for AUD only, primarily noting reductions in PTSD yet minimal effects on AUD outcomes (van Dam et al., 2012; Simpson et al., 2021; Roberts et al., 2022). The first meta-analysis of individual-level patient data of comorbid PTSD and SUD treatment trials has offered uniquely expanded and novel insights (Hien, in press). Applying specialized techniques to address limitations such as measurement and selection biases in summary data integration, the analysis harmonized data from 36 behavioral, pharmacological, and combination treatment studies ($n = 4046$), compared each intervention to treatment-as-usual, and provided head-to-head comparisons (i.e., comparative efficacy) across treatment types. Treatment types included trauma-focused therapy, integrated behavioral PTSD + AUD treatment, pharmacotherapy for PTSD (e.g., sertraline, paroxetine), pharmacotherapy for SUD (e.g., naltrexone, methadone), and stand-alone behavioral AUD therapy and treatment-as-usual, non-manualized care. Findings suggest that trauma-focused therapies (e.g., prolonged exposure and cognitive processing therapy) when combined with AUD pharmacotherapy (e.g., naltrexone and zonisamide) provide optimally effective care for comorbid PTSD and AUD, showing the largest comparative effect size (CES) against treatment-as-usual ($d = -.92$ for PTSD, $d = -1.10$ for alcohol use at end-of-treatment). Findings also highlight the relative efficacy of several other tested approaches. For example, also evidencing large CES were stand-alone trauma-focused interventions ($d = -.24$ for PTSD, $d = -.45$ for alcohol use at end-of-treatment), stand-alone pharmacotherapy for AUD ($d = -.41$ for PTSD, $d = -.83$ for alcohol use at end-of-treatment), and integrated trauma-focused and AUD behavioral therapies ($d = -.47$ for PTSD, $d = -.42$ for alcohol use at end-of-treatment). Improvements in PTSD + AUD symptoms were found for almost all evaluated treatment types, which is consistent with a prior review's conclusion of "no wrong door" for PTSD + AUD treatment (Simpson et al., 2021). Because study inclusion required a formal diagnosis of AUD or SUD, relevance of these findings to the treatment of alcohol misuse remains unknown. Critically, all 36 trials included in this meta-analysis were conducted in either the USA, Germany, or Australia, leaving the question of their applicability to LMIC contexts unanswered.

## Treatment of PTSD + AUD in LMICs

To our knowledge, no empirical studies have explored the comparative efficacy of concurrent PTSD + AUD treatment approaches in LMICs. Compounding the lack of evidence for these integrated treatment approaches, there is also sparse evidence from LMIC settings for the targeted treatment of alcohol use problems in samples with PTSD or of PTSD in alcohol-using samples.

Systematic reviews indicate that trauma-focused CBT, EMDR, and narrative exposure therapy (which embeds exposure-based CBT in an autobiographical narrative) are effective in reducing PTSD in LMIC contexts of mass violence or humanitarian crisis (Morina et al., 2016; Purgato et al., 2018). However, the effect of these treatments on concurrent AUD has not been evaluated. Similarly, while there is evidence that AUD treatments such as CBT, motivational interviewing (MI), and structured brief interventions are effective in reducing alcohol use and alcohol-related harm in LMIC settings (Preusse et al., 2020; Ghosh et al., 2022; Staton et al., 2022), effects on concurrent PTSD have seldom been reported. It is also unclear whether PTSD + AUD acts as a moderator on treatment outcomes. In sum, samples with PTSD and samples with AUD have largely been conceptualized as distinct clinical populations in treatment research in LMICs to date. This is concerning given the increased risk of PTSD + AUD that has been reported in both at-risk and general populations in LMIC settings.

## Transdiagnostic approaches

Given high rates of PTSD + AUD globally, another intervention option that has been gaining in popularity is the use of a *transdiagnostic* therapy approach. Transdiagnostic interventions were developed to address a broad range of commonly comorbid mental and behavioral health problems more effectively and efficiently. This approach differs from concurrent treatment in that it integrates evidence-based therapeutic techniques common to treatments for depression, anxiety, and other mental health problems and aims to address shared mechanisms causing these problems, such as cognitive and behavioral avoidance.

There are at least three reasons a transdiagnostic approach in LMICs may be preferable to traditional concurrent approaches. Transdiagnostic treatments are intrinsically designed to (1) be delivered by a single provider; (2) address additional comorbidities beyond PTSD + AUD; and (3) address a range of symptom severity, such that they do not require a formal diagnosis. The latter strength is especially helpful in a LMIC context where comprehensive assessment and diagnosis may not be feasible (Murray et al., 2014; Kane et al., 2018). Two transdiagnostic approaches have been developed and tested in LMICs: the Common Elements Treatment Approach (CETA) and Problem Management Plus (PM+; Dawson et al., 2015). In both cases, these treatments were not initially designed to address AUD but have since been adapted to do so. Recently, an integrated trauma-focused substance use treatment was also developed in South Africa (Myers et al., 2019).

CETA was designed as a multi-problem, flexible transdiagnostic intervention that could be delivered by lay counselors to address a range of mental and behavioral health problems (Murray et al., 2014). It was originally designed to focus on common mental health problems of PTSD, depression, anxiety, behavioral issues (for children and adolescents), and impaired relationship and psychosocial functioning. CETA was based on transdiagnostic treatments developed in HICs (Chorpita and Daleiden, 2009; Farchione et al., 2012; Weisz et al., 2012), and treatment components were based on CBTs. Across six LMIC-based randomized-controlled trials (RCTs) of CETA, the treatment has demonstrated robust evidence for treating PTSD among trauma-affected populations (Bolton et al., 2014; Weiss et al., 2015; Bonilla-Escobar et al., 2018; Murray et al., 2020; Bogdanov et al., 2021; Kane et al., 2022).

Bolton et al. (2014) first attempted to address co-occurring unhealthy alcohol use and trauma with CETA in a RCT conducted among Burmese refugees in Thailand. This version of CETA added

MI aimed at reducing unhealthy alcohol use to the existing intervention. Although the trial demonstrated strong effects of CETA in treating PTSD, there were no impacts on alcohol misuse. Methodological limitations, including the quality of the alcohol screening measure and a small sample size of individuals who reported alcohol misuse, may have contributed to the null findings. However, there was also a concern that MI components were challenging for lay counselors to deliver with fidelity.

Following the Thailand trial, the alcohol/substance use reduction element of CETA was revised as a CBT-based substance use reduction and relapse prevention component (Henggeler et al., 2002; Danielson et al., 2012; Kane et al., 2017). In two recent CETA trials conducted in Zambia, evidence suggested effectiveness of this modified CETA protocol in treating PTSD + AUD. In the first trial, CETA significantly reduced alcohol use among both women and men in families with ongoing interpersonal violence and among whom PTSD was highly prevalent (Murray et al., 2020). In the second trial, CETA reduced unhealthy drinking among people living with HIV, and a subgroup analysis in the trial indicated that the effect size for CETA was largest among those who had unhealthy alcohol use and mental health comorbidities, including PTSD (Kane et al., 2022).

PM+ was developed by the World Health Organization as a *low-intensity intervention*, meaning that it was designed to be less time- and resource-consuming when compared to formal, high intensity psychological treatments delivered by mental health professionals and thus increases feasibility and sustainability potential in LMICs (Dawson et al., 2015). PM+ is a transdiagnostic approach designed to be delivered by lay providers to address a range of common mental health (e.g., depression, anxiety, stress, or grief) and practical problems (e.g., unemployment, interpersonal conflict). Compared to CETA, PM+ has a stronger focus on behavioral, as opposed to cognitive, therapeutic techniques based on the rationale that lay providers would be able to implement behavioral strategies more easily than cognitive interventions. Because it is a less intensive transdiagnostic intervention than CETA, PM+ is considered inappropriate for more severe mental health problems (Dawson et al., 2015; World Health Organization, 2018). PM+ has been tested in three RCTs in LMICs (Rahman et al., 2016, 2019; Bryant et al., 2017). All three trials found effectiveness of PM+ in addressing the stated primary outcomes (i.e., reductions in psychological distress, depression, anxiety). Efficacy outcomes of PM+ on PTSD have varied, and alcohol use was not measured. However, there is currently an effort underway to develop and test an alcohol-specific component to PM+ (PM + A) in trials in Ukraine and Uganda (Fuhr et al., 2021).

In South Africa, Myers et al. (2019) developed a trauma-informed intervention called Women's Health Coop to address the interrelated problems of substance use, traumatic stress, and sexual risk behaviors in young women from low-income communities. Women's Health Coop is a 6-session, group-based treatment that integrates feminist empowerment theory and CBT components. A single-arm feasibility study with 60 participants reported significant reductions in alcohol and other substance use, PTSD symptoms, depression symptoms, and sexual risk behaviors at 3 months post-treatment (Myers et al., 2019).

## Discussion

The aim of the present review was to examine the literature to date on the prevalence, impact, theories of etiology, and treatment approaches to addressing PTSD + AUD in LMICs, which have historically gone underrepresented in PTSD + AUD research. Information on PTSD + AUD prevalence, phenomenology, symptom severity, scope and range of consequent functional impairment, and mechanisms of maintenance in LMICs is currently inconclusive, with limitations stemming from heterogeneity in the evidence derived from various studies, as well as sampling and methodological variation. There appears to be some evidence that both PTSD and AUD occur less frequently in LMICs than HICs, but that symptoms and consequences are more severe when they do occur. The reasons for these findings are not well understood and do not address prevalence or severity rates of PTSD + AUD specifically (Loring, 2014; Koenen et al., 2017). The majority of the literature on PTSD + AUD to date supports a self-medication-based relationship between PTSD symptoms and alcohol use, but there is little research evaluating this theory or culturally informed alternatives in LMICs. Moreover, there is a need for more research that clearly evaluates PTSD + AUD within a societal and cultural context to help elucidate possible differences in risk and protective factors across populations.

The present review also examined the existing literature on treatment, including approaches that address AUD and PTSD individually, approaches that directly address concurrently PTSD + AUD, and transdiagnostic approaches that might address AUD and/or PTSD among other mental and/or behavioral health concerns. PTSD + AUD treatment has largely been examined in HICs, where the evidence base has moved from sequential to concurrent treatment models. However, PTSD and alcohol treatment studies conducted in LMICs are largely siloed, with studies failing to address PTSD + AUD. Though several evidence-based treatments developed in HICs, including intensive trauma-focused therapies, have been successfully implemented in LMICs, there are barriers to broad dissemination and implementation efforts in these settings (i.e., time and resource limitations, the fact that treatments were designed to be implemented by trained specialists rather than lay professionals). Novel interventions being tested in LMICs, such as transdiagnostic treatment approaches that target common mechanisms of psychopathology and can be implemented by a single provider, have potential to provide new avenues for intervention on PTSD + AUD. As of this writing, only CETA has demonstrated efficacy in randomized trials, and other concurrent models have yet to be tested for their effects on PTSD + AUD. Moreover, research on the efficacy of PTSD + AUD treatment delivery by lay professionals in LMICs has potential to expand access to PTSD + AUD care globally, in both LMICs and HICs. Results of this review also highlight that a syndemics approach to alcohol treatment, in which addressing AUD was embedded into a broader public health interventions for other diseases highly relevant to LMIC contexts, such as HIV, shows promise in reducing heavy drinking and related consequences. However, there is a need for studies of these interventions to test whether they also improve PTSD.

One major challenge in understanding relationships between trauma exposure, PTSD, and alcohol misuse is the paucity of data globally, despite evidence that both PTSD and AUD have a large impact on population health that disproportionally affects those in LMICs. This lack of representation means that many of our assumptions as a field regarding how these disorders are interrelated are based on a relatively small and globally non-representative sample. Research examining potential reasons for geographic and cultural variations in symptom presentation and comorbidity may provide the groundwork for development for new prevention and intervention strategies.

**Open peer review.** To view the open peer review materials for this article, please visit http://doi.org/10.1017/gmh.2022.63.

**Data availability statement.** Not applicable to this narrative review.

**Acknowledgements.** This article was supported by grants from the NIMHD (R01MD011574, PIs: Pearson/Kaysen), NIMH (T32MH019938, PIs: Schatzberg/Manber; K01MH129572, PI: Greene), and NIAAA (K01AA026523, PI: Kane).

**Author contributions.** Drs. Debra Kaysen was responsible for the conception and design of the this review. All authors contributed to material preparation, literature reviews, and manuscript preparation. All authors read and approved the final manuscript.

**Financial support.** This work was supported by the National Institute of Mental Health (grant number T32MH019938); the National Institute on Alcohol Abuse and Alcoholism (grant number K01AA026523); and the National Institute on Minority Health and Health Disparities (grant number R01MD011574).

**Competing interest.** We have no known conflicts of interest to disclose.

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
