## [Reviewer Report]

September 1, 2022

Dear Dr. Belkin, 

We have uploaded a manuscript entitled, “Comorbid PTSD and Alcohol Use Disorder in Low- and Middle-Income Countries: A Narrative Review” for your review for publication in Global Mental Health. 

This manuscript seeks to review the literature to data on the co-occurrence of PTSD and Alcohol Use Disorder in low- and middle-income countries, which have long been neglected in the development of our epidemiological and theoretical understanding of the comorbidity, limiting the development of globally-relevant treatments. 

The purpose of this review is to identify current state of the scientific literature on the comorbidity between PTSD and Alcohol Use Disorder stemming from limited investigations of these constructs in Low- and Middle-Income Countries (LMICs) around the world. Discussion and conclusions focus specifically on the limitations in our understanding of comorbid PTSD and Alcohol Use Disorder based on the historic neglect to investigate these constructs in LMICs. Improvement/expansion of our understanding of the global incidence, consequences, and treatment best practices of comorbid PTSD and Alcohol Use Disorder will require the inclusion of more data from LMICs. As part of our review we attempted to include studies using a global lens, rather than focusing on a specific region. 

We hope that you will give our manuscript serious consideration for publication in your journal, as we believe it is timely and will make a substantive contribution to an emerging literature on this topic.

Thank you for your consideration of our work and we look forward to hearing from you. 

Sincerely,

Debra Kaysen, Ph.D., ABPP

Professor

Department of Psychiatry and Behavioral Sciences

Stanford University

Email: dkaysen@stanford.edu

---

## [Reviewer Report]

*Comments to Author*: The authors present a narrative review on the co-occurrence of posttraumatic stress disorder (PTSD) and alcohol use disorders (AUD) in low- and middle-income countries (LMIC). Each of the disorders alone and especially their comorbidity (PTSD+AUD) are of considerable importance from a public health perspective. While numerous findings on their epidemiology, underlying mechanisms and treatment of PTSD+AUD are available from high income countries (HIC), this is not the case for LMIC so far. In their manuscript, the authors contrast the current state of knowledge from HIC with the (often very limited) findings from LMIC. They also address methodological aspects and make recommendations for future studies. In my view, the topic is absolutely timely, the manuscript is very well written, and I only have a few comments.

Page 4, 1st line (“… and more than other anxiety disorders…”): Since PTSD is no longer grouped with the anxiety disorders in DSM and ICD, the authors could consider to change this sentence accordingly.

Page 5, 2nd paragraph, 2nd last sentence (“This increased risk…”): “in” should be deleted.

Page 12, last sentence of 1st paragraph (“To our knowledge…”): As this sentence deals with treatment approaches in LMIC, it could also be integrated in the next section, not the section on HIC.

Page 16 (“There appears to be some evidence that both PTSD and AUD occur less frequently in LMICs than high-income countries, but that symptoms and consequences are more severe when they to occur”): I may have missed out the respective information, but these facts seem to be mentioned for the first time here? If so, I would suggest to insert some references.

---

## [Reviewer Report]

*Comments to Author*: I thought this was a really interesting and informative review article and it provides a great overview of this field. I also liked the fact that it outlines the current treatment options for people with co-occurring problems which hasn't been provided in may other papers.

I have outlined some suggestions and comments which I thought could improve the readability of the paper.

Page 4 - para 4 - It is discussed that there is likely to be a greater burden of harm from comorbidity in LMICs than in HICs, but I wanted to understand more what types of harms this is referring to and if that is interpreted in the context of these countries, which for example, will have fewer healthcare resources than HICs.

Page 5 - in discussion of the epidemiology and definitions of AUD, I think it could also be helpful to mention issues around assessment and the fact that some of the standardised assessments e.g. AUDIT may not be as culturally appropriate for some countries, particularly where there may be a higher level of not-drinking or if alcohol is prohibited.

Across pages 5 and 6 - studies which have assessed co-occurrence in LMICs are discussed, but the reporting is quite broad and I wondered if it might be helpful to refer more specifically to the level of co-occurrence that these studies identified, for example, as a %.

Page 7 - section on the theoretical models - I thought that the authors may want to refer back to a previous point raised that many of the studies in LMICs have focused on specific populations who have experienced trauma, and how this may impact on the understanding of the theoretical models and whether population level studies may be required to understand this further.

Page 8 - para 1 - I thought that a definition of 'micro-longitudinal' should be provided.

Page 11- The overview of the meta analysis findings are really interesting and I wondered if a bit more detail could be provided on the specific types of treatment, or whether this could be summarised in a table? As I think this is valuable information in understanding the effectiveness of different types of therapies. And could the effect sizes be provided for the other treatments in addition to the combined therapy.

Pages 14-15 - Across the discussion of transdiagnostic approaches I would have liked to hear more about why the interventions haven't always had the expected impact on alcohol use and the reasons why that might be.

Discussion - I thought that the above point could be continued in the discussion around what approach will be best in LMICs and why some of the more intensive trauma focused therapies which have been tested in HICs may not be feasible, or what the requirements may be in order for these to be rolled out more widely in other contexts.

---

## [Reviewer Report]

*Comments to Author*: This is a well-written, cogent, and concise review detailing major studies in the field that assess comorbid AUD and PTSD in low-and-middle-income settings. It sheds light on methodological challenges involved in comparing between studies that use distinct definitions and measurement tools. A major strength of this review is the consideration of how evidence form low-and-middle income countries fits within existing theories; additionally, the authors’ call to conduct more studies exploring local cultural explanations and theories that have not yet been considered is well articulated.

---

## [Reviewer Report]

Thank you for the useful reviews of our original manuscript entitled “Comorbid PTSD and Alcohol Use Disorder in Low- and Middle-Income Countries: A Narrative Review” submitted to Global Mental Health for consideration. We appreciate the reviewers’ kind remarks regarding the timeliness and importance of this review. We also appreciate the effort the reviewers put into the extremely helpful reviews. We have attempted to address all reviewer comments. We believe the manuscript is substantially improved as a result of this feedback. 

All co-authors contributed substantively to the paper and have approved this submission. Authors have no conflicts of interest to declare pertinent to this submission.

We very much hope that our work is of interest to you and your readership and that the revised manuscript is a good match for your journal.

Sincerely,

Debra Kaysen, Ph.D., ABPP

Professor

Chief, Division of Public Mental Health & Population Sciences

Department of Psychiatry

Stanford University School of Medicine